# THE TELEPHONE GAME: EVALUATING SEMANTIC DRIFT IN UNIFIED MODELS

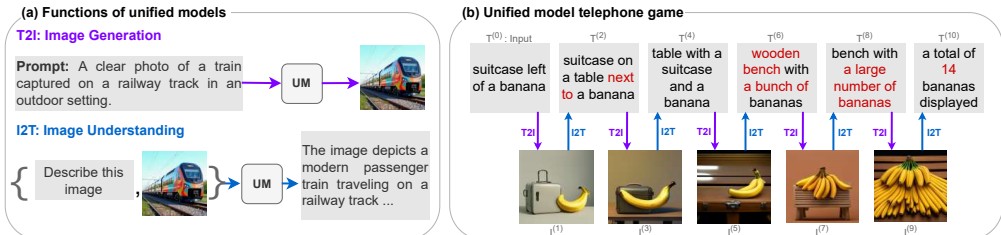

Figure 1: (a) **Unified models** possess both image generation and image understanding functionalities. (b) **Telephone Game**: Here, the unified model starts from a textual prompt $T^{(0)}$ about a suitcase and a banana. After successive T2I & I2T steps we observe in the $5^{th}$ generation image, $I^{(5)}$, the model fails to generate a convincing suitcase. Subsequently the suitcase disappeared from future generations. Also, $I^{(5)}$ has two bananas instead of one, which culminated in lots of bananas.

## ABSTRACT

Employing a single, unified model (UM) for both visual understanding (image-to-text: I2T) and visual generation (text-to-image: T2I) has opened a new direction in Visual Language Model (VLM) research. While UMs can also support broader unimodal tasks (e.g., text-to-text, image-to-image), we focus on the core cross-modal pair T2I and I2T. Existing evaluations benchmarks consider these capabilities in isolation: FID and GenEval for T2I, and benchmarks such as MME, MMBench for I2T. These isolated single-pass metrics do not reveal cross-consistency: whether a model that "understands" a concept can also "render" it, nor whether semantic meaning is preserved when cycling between image and text modalities. To address this, we introduce the Semantic Drift Protocol (SDP) for Unified Models, a cyclic evaluation protocol that alternates I2T and T2I over multiple generations to quantify semantic drift. We propose two metrics: (i) Mean Cumulative Drift (MCD), an embedding-based measure of overall semantic loss; and (ii) Multi-Generation GenEval (MGG), an object-level compliance score extending GenEval. To assess generalization beyond COCO dataset, which is widely used in training; we create a new benchmark Nocaps+Docci400, sampled from NoCaps and DOCCI and evaluate on seven recent models. SDP reveals substantial variation in cross-modal stability: some models like BAGEL maintain semantics over many alternations, whereas others like Vila-u drift quickly despite strong single-pass scores. Our results highlight SDP as a necessary complement to standard I2T and T2I evaluations.

## 1 INTRODUCTION

Multimodal Unified Models (UMs) combine visual understanding and generation within a single framework, enabling a wide range of unimodal tasks (e.g., text-to-text, image-to-image) as well as cross-modal tasks (e.g., image-to-text, text-to-image). By sharing representations across modalities, UMs can demonstrate interesting emerging capabilities such as intelligent photo editing, e.g. BAGEL Deng et al. (2025). Despite rapid model progress, UM evaluation remains fragmented. Existing metrics assess image understanding and image generation in isolation; e.g., MME, MM-Bench, POPE, VQA Fu et al. (2024); Liu et al. (2024); Li et al. (2023); Agrawal et al. (2016) are used for evaluating understanding (I2T), and Inception score, CLIPScore, FID, GenEval Radford et al. (2016); Heusel et al. (2017); Ghosh et al. (2023) are used for evaluating image synthesis (T2I),

while overlooking the retention of important information during `T2I` or `I2T` multi-turn conversion. In other words, current single-pass metrics do not assess the retention of entities, attributes, relations, and counts under alternating `I2T` ↔ `T2I` conversions. We defer unimodal tasks and center our analysis on `I2T` and `T2I` tasks as the potential for semantic divergence and its impact on real use is most pronounced on the cross-modal tasks.

We begin by formalizing two key notions, "semantic-drift" and " cross-consistency". Semantic drift is the loss or distortion of meaning that accumulates when an input is repeatedly transformed across modalities via `T2I` and `I2T`. Essentially, this drift can be defined as the changes in the core semantic content (e.g: objects count, color, attribute relations, spatial position) that occur when a model repeatedly applies its own I2T ↔ T2I transformations. On the other hand, cross-consistency refers to the overlap between what a model can generate as images from text and what it can faithfully understand from images as text. Much like the popular children's game called *Telephone Game*, where a whispered message drifts in meaning as it passes from person to person, UMs tend to lose or distort semantic meaning when cycling between text and image representations as shown in Fig. 1(b). Starting from a textual prompt: "a suitcase left of a banana", the model produces an image $I^{(1)}$ correctly, which is then captioned (`I2T`) to form the next prompt $T^{(2)}$, and so on. Although each individual step can look plausible in isolation, se-

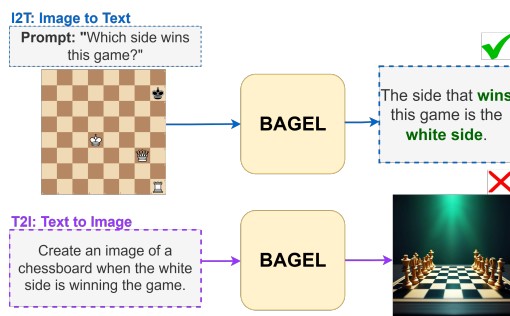

Figure 2: An example of cross-consistency in the BAGEL unified model. Given an image of a chess board along with a question (top), BAGEL performs `I2T`, correctly answering "white side wins". By creating another caption for the `T2I` prompt (bottom), BAGEL should generate a chess board image consistent with the same semantic predicate (white winning side). However, the model generates a generic, mismatched chessboard image. This exposes a unified model inconsistency: BAGEL's correct visual reasoning (`I2T`) does not carry over to generation (`T2I`) for the concept "winning side in chess".

mantic drift accumulates across the cycles: by generation 5, the image has changed drastically. Notably, a model may score well on isolated single-pass `I2T` or `T2I` metrics, while still exhibiting these cross-modal inconsistencies, which the current metrics fail to capture. The concept of cross-consistency is illustrated in Fig. 2, where even state-of-the-art unified models like BAGEL Deng et al. (2025) can correctly reason about a chessboard image in `I2T` identifying that "the white side wins", yet fail to produce a faithful `T2I` image of the same winning scenario.

There are several ways to evaluate a model's image generation capabilities. For example, ClipScore Hessel et al. (2022) uses clip embeddings to measure semantic alignment of the prompt with generated images. However, it strongly relies on clip embeddings, which may not always be reflective with human perceptions Ghosh et al. (2023). Fréchet Inception Distance (FID) Heusel et al. (2017) measures the distributional similarity between the generated images and real images, but ignores the generated image's faithfulness to the input prompt. A model that ignores the input text and produces high-quality, yet off-prompt images can still score well Ghosh et al. (2023). GenEval Ghosh et al. (2023) improves on prompt alignment by checking object and relation-level compliance with detection models, however, by design, does not assess overall visual quality or realism, and like FID, remains a single-pass measure. A similar limitation is observed in the image-understanding benchmarks, such as MME and MMBench Fu et al. (2024); Liu et al. (2024) which assess `I2T` skills in isolation, without testing whether the model's understanding capability aligns with its generation capability.

To address this gap, we evaluate unified models cross-consistency and drift in single- and multi-turn settings respectively. In the single-pass setting (one-step `I2T` and `T2I` on paired image–caption data), we perform a human cross-consistency study to judge consistency between model outputs relative to its inputs. In multi-pass, we propose the Semantic Drift Protocol for Unified Models (`SDP`), a cyclic evaluation protocol designed to quantify how well UMs preserve semantic meaning under repeated `T2I` and `I2T` conversions. Starting from an initial input $T^{(0)}$ (text) or $I^{(0)}$ (image), the model alternates `T2I` or `I2T` to produce a sequence $\{I^{(g)}, T^{(g)}\}$, where $g$ denotes generation step. At each generation $g$, `SDP` measures semantic similarity back to the initial input and across steps, capturing drift directions and exposing misalignment between a model's understand-

ing and generation spaces. We employ CLIP Radford et al. (2021), DINO Caron et al. (2021), and MPNet Song et al. (2020) embeddings for text–image, image–image, and text–text comparisons, respectively. For rigorous testing, we design two different metrics: Mean Cumulative Drift (`MCD`), and Multi-Generation Geneval (`MGG`). In `MCD`, we use raw embedding distance scores to quantify cumulative information retention, and `MGG` extends the GenEval benchmark for multiple generations. We propose a new benchmark dataset `Nocaps+Docci400`, sampling 200 image-text pairs from NoCaps Agrawal et al. (2019) and 200 image-text pairs DOCCI Onoe et al. (2024) datasets. These two datasets were selected for their novel objects and fine-grained visual details that better probe generalization. We benchmark 7 recent models spanning shared-weight, partially shared, and decoupled architectures, to analyze how architectural design choices influence semantic stability. Further, to validate the proposed embedding metrics, we also ask humans to rank the model outputs: we conduct a human study in which annotators score the fidelity of each output relative to its input and provide comparative rankings across multiple model outputs. The fidelity scores indicate the degree to which inconsistencies are present, while the rankings establish relative model performance according to human judgment.

Our experiments reveal substantial variation in semantic drift behavior across models. For example, BAGEL Deng et al. (2025) maintains strong semantic fidelity across multiple generation cycles, whereas models like Vila-U Wu et al. (2025) and Janus Wu et al. (2024) degrade rapidly, exposing weaker coupling between their visual understanding and visual generation capabilities despite competitive single-pass metrics. These findings underscore the need to move beyond isolated `I2T` or `T2I` metrics and toward evaluations that directly measure cross-consistency.

Our contributions are summarized as follows:

- We formalize the cross-consistency and semantic drift problem, showing that single-pass metrics cannot expose gaps between a model's understanding and generation capabilities.
- We propose the Semantic Drift Protocol (`SDP`), which jointly evaluates `I2T` and `T2I` over multiple transitions to track semantic preservation.
- We extend GenEval Ghosh et al. (2023) to a multi-generation setting, which amplifies observable performance differences between models.
- We conduct a human study to determine cross-consistency in existing models and provide a comparative ranking.

## 2 UNIFIED MODELS

Unified models employ visual and textual modalities as both input and output. The motivation is that these universal models facilitate richer semantic interoperability among the two tasks, `I2T` and `T2I`.

While most prior works focus on building a single model for both tasks, we propose a broader categorization that encompasses unified models as well as models that can emulate unified behavior.

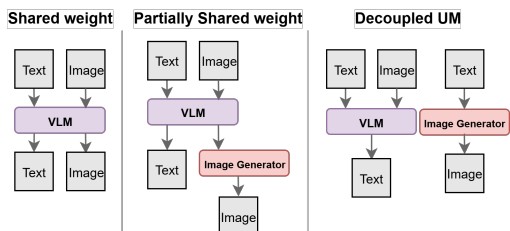

**Shared-Weights Unified Models** This category has received the most attention in recent research. These models leverage a single model, typically a transformer decoder, to perform a wide spectrum of unimodal and cross-modal tasks, with `T2I` and `I2T` generation being prominent examples. The encoder component can vary where some models employ a shared visual encoder across tasks, while others use distinct encoders for generation and understanding. In our experiments, we use 5 such models: BAGEL Deng et al. (2025), Janus 1.3B Wu et al. (2024), Janus Pro 7B Wu et al. (2024), Show-o Xie et al. (2024), and Vila-u Wu et al. (2025).

Figure 3: On the left, a single model handles both understanding and generation. In the middle, the architecture partially shares weights, with a decoder capable of generating text and visual features, the latter is passed to another image generation model. On the right, the understanding and generation processes are fully decoupled, using separate models for each task.

**Partially Shared Models** Models in this category retain a degree of parameter sharing, while delegating specific responsibilities to task-specific modules. This design allows more flexibility in handling modality-specific complexities while preserving shared knowledge across tasks. We use *Blip-3o* Chen et al. (2025) which incorporates a dedicated diffusion model for image generation.

**Decoupled Models** Models in the third category are formed by constructing a unified pipeline by composing independently trained models, which in tandem can emulate unified behavior. The example we have used is pairing a VLM like *LLaVA*Liu et al. (2023) for `I2T` with a *Stable Diffusion* Podell et al. (2023) model for `T2I`. This setup enables task interoperability without requiring joint training or weight sharing.

## 3 Semantic Drift Evaluation

We propose a cyclic evaluation Protocol `SDP` which provides three different metrics to measures how well a unified model preserves semantic fidelity when alternating between `I2T` and `T2I`. `SDP` proposes to evaluate on multi-generation cycles to provide quantitative measures of semantic drift. In this setting, we treat the $\mathcal{UM}$ as a model composed of at least two functionalities. **Image Generation**: $\mathcal{UM}_{\texttt{T2I}} : \mathcal{T} \to \mathcal{I}$, which synthesizes an image given a textual description. **Image Understanding (`I2T`)**: $\mathcal{UM}_{\texttt{I2T}} : \mathcal{I} \to \mathcal{T}$, which generates a textual description from a given image. Here, $\mathcal{T}$ denotes the set of all possible text representations (e.g., captions, instructions), and $\mathcal{I}$ denotes the set of all possible image representations.

Let $\mathcal{D} = \{(I_i, T_i)\}_{i=1}^N$ represent a dataset of $N$ paired samples, where each $I_i \in \mathcal{I}$ and each $T_i \in \mathcal{T}$ is its corresponding caption. A *generation step* is defined as the application of either $\mathcal{UM}_{\texttt{T2I}}$ or $\mathcal{UM}_{\texttt{I2T}}$ to transform an input from one modality into the other. We define alternating chains of length $G$ starting from either text or image. Let $g \in \{0, 1, \dots, G\}$ be the generation step index. Then similar to the chains defined in Bahng et al. (2025), we consider two experimental setups depending on the initial modality:

- **Text-First-Chain**: Starting from $T^{(0)}$, each step applies `T2I` then `I2T`:

$$T^{(0)} \xrightarrow{\texttt{T2I}} I^{(1)} \xrightarrow{\texttt{I2T}} T^{(2)} \xrightarrow{\texttt{T2I}} I^{(3)} \cdots$$

  Here, similarity can be measured from initial text against later texts or images, giving the distance mappings $\{\text{text} \to \text{text}, \text{text} \to \text{image}\}$.

- **Image-First-Chain**: Starting from $I^{(0)}$, each step applies `I2T` then `T2I`:

$$I^{(0)} \xrightarrow{\texttt{I2T}} T^{(1)} \xrightarrow{\texttt{T2I}} I^{(2)} \xrightarrow{\texttt{I2T}} T^{(3)} \cdots$$

  Here, similarity can be measured from initial image against later images or texts, giving the distance mappings $\{\text{image} \to \text{image}, \text{image} \to \text{text}\}$.

Depending on the modality of initial input and the modality considered for distance calculation, we define a set of distance mappings, $\Delta = \{\text{text} \to \text{text}, \text{image} \to \text{text}, \text{text} \to \text{image}, \text{image} \to \text{image}\}$.

The intuition for `SDP` is that a semantically consistent model will preserve the core meaning of the original content across many generations of alternating `T2I` and `I2T`; A weaker model will drift away from the original meaning more quickly. To systematically measure this degradation, in our protocol we propose two distinct metrics. `MCD` provides a holistic measure of drift based on embedding similarity. On the other hand, `MGG` grounds the evaluation in object-level fidelity by extending the GenEval benchmark across multiple generations.

### 3.1 MCD: Mean Cumulative Drift

`MCD` measures how much meaning a model can retain after multiple `T2I` and `I2T` cycles. To obtain this metric we compare the input with the output of later generations using embedding based similarity scores. For any dataset that has text-image pairs, we can construct two separate chains (Text-First and Image-First chains). Then, for each distance mapping $\delta \in \Delta$ we obtain a sequence

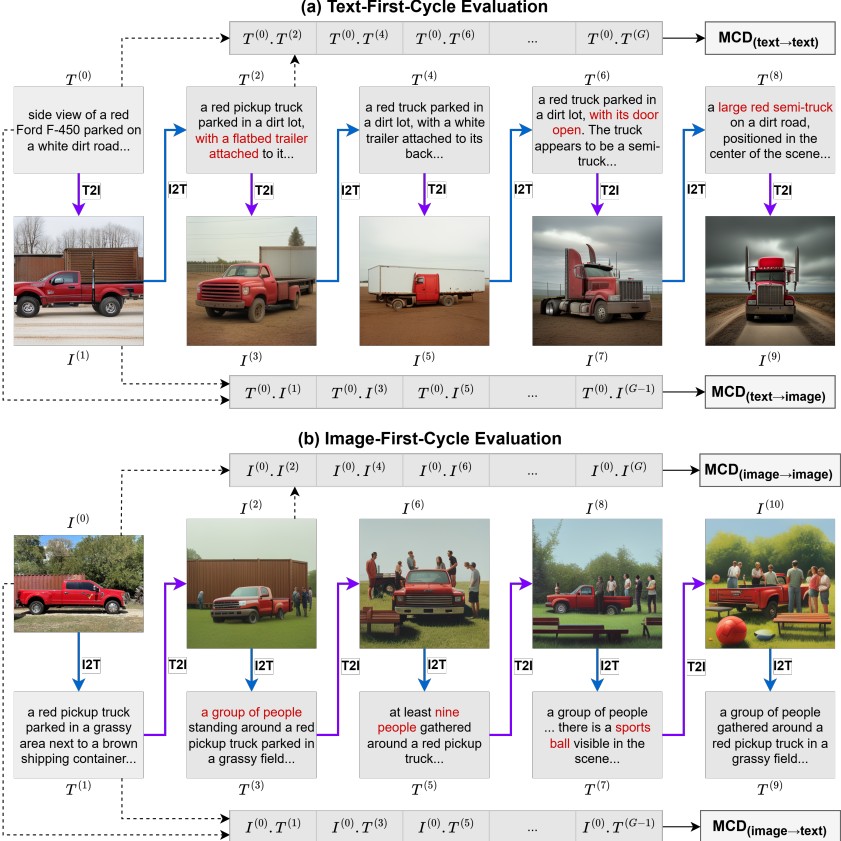

Figure 4: Semantic Drift Protocol (`SDP`). We alternate between text-to-image (`T2I`) and image-to-text (`I2T`) generations in two setups: Text-First-Chain (a) and Image-First-Chain (b). Blue arrows denote `I2T`; purple arrows denote `T2I`; dashed black arrows indicate similarities computed back to the initial input in both same- and cross-modality directions used for `MCD`. Across generations, concepts drift despite plausible single steps: a "red F-450 truck" evolves into a semi-truck with changing attachments and positions; in the image-first chain, group size inflates and new objects (e.g., a sports ball) appear. The proposed cyclic evaluation reveals cross-modal concept drift that single-pass metrics overlook, enabling direct comparison of unified model's semantic stability.

of distance scores across the generations. We then average the sequences at every generation along the entire dataset $\mathcal{D}$,

$$S_\delta(g) = \frac{1}{|\mathcal{D}|} \sum_{d \in \mathcal{D}} \text{sim}\big(inp_d, M_{d,\delta}^{(g)}\big) \tag{1}$$

where $S_\delta(g)$ is the average similarity at generation $g$ for distance mapping $\delta$, $M_{d,\delta}^{(g)}$ is the generated text or image at generation $g$. Here, $sim$ denotes distance function using one of the embedding models (CLIP, DINO, or MPNet). To get overall drift, we compute mean across generations $S_\delta(g)$,

$$\text{MCD}_\delta = \frac{1}{G} \sum_{g=1}^{G} \big(S_\delta(g)\big), \tag{2}$$

where $\text{MCD}_\delta$ is a single integer denoting mean cumulative drift for a given distance mapping. To compute across all mappings, we compute mean across all distance mappings to get $\text{MCD}_{avg}$. A higher `MCD` means the chain retains its semantic meaning more consistently across generations, while a lower value indicates higher drift.

## 3.2 MGG: MULTI-GENERATION GENEVAL

To complement embedding-based similarities with object-level fidelity, we further extend GenEval Ghosh et al. (2023) to our proposed multi-generation setting. The existing Geneval protocol Ghosh et al. (2023) is designed to assess text-to-image fidelity across multiple dimensions of

quality. These dimensions include *single_object*, *two_object*, *counting*, *colors*, and *positions*, and *attributes_binding*. For each task, GenEval proposes a diverse set of prompts such as "a photo of a/an [COLOR] [OBJECT]". Once a model has generated images for all the prompts, GenEval uses a pre-trained object detection model to detect and localize objects in the generated images. This process allows us to calculate the accuracy of the model for each task. An average of the task level accuracies is then denoted by GenEval overall accuracy. We build on the existing benchmark by incorporating the GenEval Rewritten dataset Chen et al. (2025), adopting the newer OwlV2 object detection model Minderer et al. (2024), and extending evaluation across multiple generations. To calculate MGG, we first calculate the GenEval scores for each generation for all tasks. Then, similar to GenEval overall accuracy, we compute the tasks scores to obtain GenEval overall accuracy for each generation. Finally, we average the generation scores to obtain the MGG score. Higher MGG scores indicate better ability to produce semantically accurate and, context-preserving outputs.

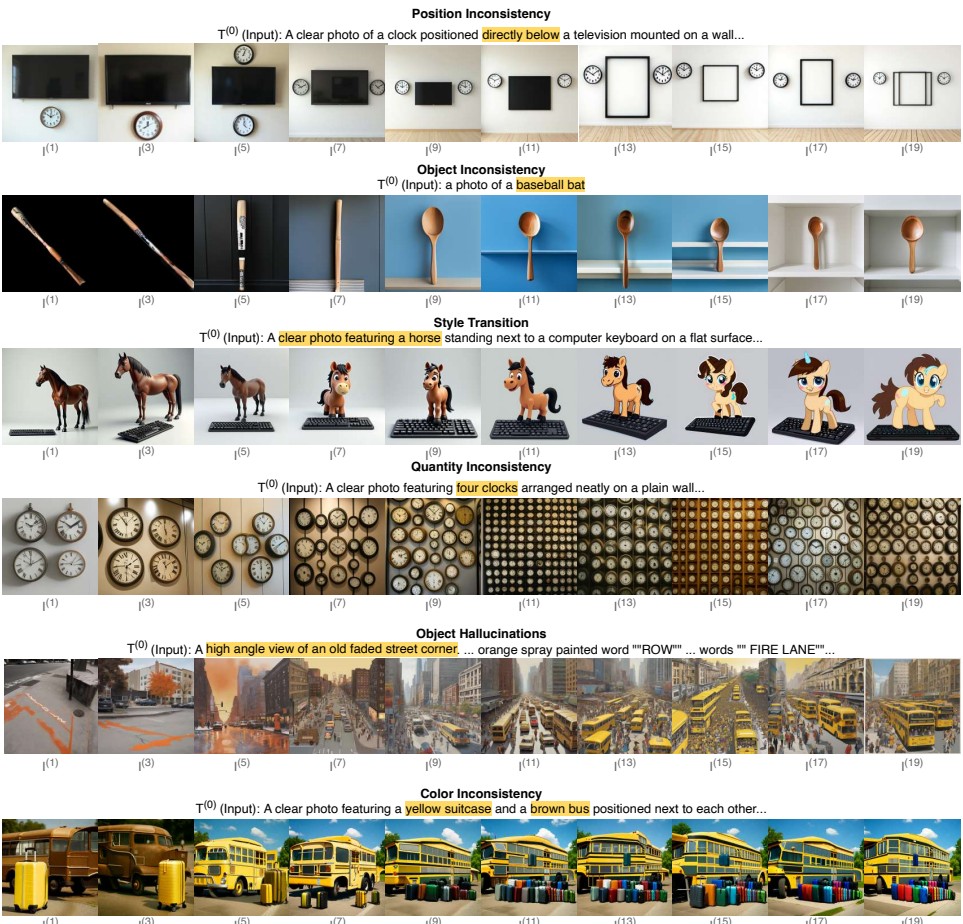

Figure 5: Information can be lost in different ways during a cyclic inference. In the first row, the model ignores the position of the clock, which is a crucial detail. In the second row, the model changes a baseball bat into a spoon. A model can also change the style from realistic to cartoon, as shown in the third row. In the fourth row the model loses count of four clocks and generates lots of clocks instead. In the fifth row a whole city is hallucinated around an empty road. In the sixth row, the model changes a brown bus into a yellow bus.

## 3.3 SINGLE-PASS HUMAN EVALUATION (CROSS-CONSISTENCY)

We complement our cyclic analysis with a single-step cross-consistency evaluation to highlight cross-modal fidelity issues, sampling 100 Text-First and 100 Image-First chains from the MCD evaluation set for a total of 200 examples. Given a ground-truth pair $(I, T)$, we first generate a caption $T^{(1)} = \text{UM}_{\text{I2T}}(I)$ via I2T and an image $I^{(1)} = \text{UM}_{\text{T2I}}(T)$ via T2I. We then assess whether

$T^{(1)}$ and $I^{(1)}$ preserve the semantics of $(I, T)$ along two axes: (a) $I \rightarrow T^{(1)}$ consistency—does $T^{(1)}$ faithfully describe $I$? and (b) $T \rightarrow I^{(1)}$ consistency—does $I^{(1)}$ depict $T$? Six human annotators participated in the study; each received a comparable workload, and every example was evaluated independently by two different annotators. Using a web interface, annotators provided two judgments per sample: a three-level fidelity score (Good, Medium, or Poor)and a ranking of model outputs based on semantic correctness relative to the original input. To ensure unbiased evaluation and prevent positional bias, model identities were masked and the output order was randomized for every instance. Each sample page contained two sections: in the **understanding section**, annotators rated and ranked captions for the input image; in the **generation section**, they rated and ranked generated images for the input text prompt. Finally, rather than averaging annotators' opinions, we treat each annotation as an independent data point, allowing us to measure consistency without collapsing individual perspectives.

## 4 EVALUATIONS & FINDINGS

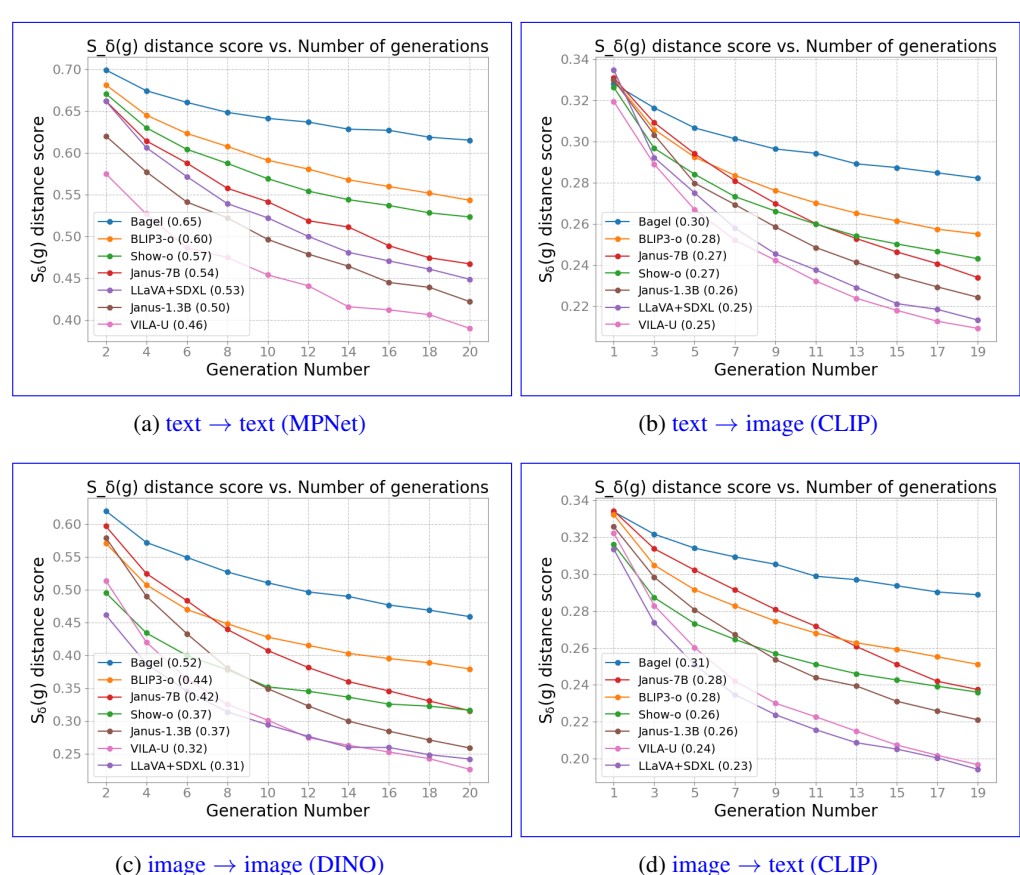

(a) text → text (MPNet)

(b) text → image (CLIP)

(c) image → image (DINO)

(d) image → text (CLIP)

Figure 6: The graph shows $S_\delta(g)$ distance scores computed using Eq. 1. Plots showing Text First (a)(b) and Image First (c)(d) chains that illustrate semantic drift across generations. The legend mentions average scores across all generations for the given distance mapping.

For embedding based semantic drift analysis (MCD), we randomly sample 200 image-text pairs from each of the two challenging vision-language datasets, Nocaps Agrawal et al. (2019) and DOCCI Onoe et al. (2024). We denote this sample dataset as Nocaps+Docci400. These corpora stress both novel objects and fine-grained details, making them well-suited to reveal drift that single-pass metrics do not capture. NoCaps introduces nearly 400 novel objects unseen in COCO and features more visually complex images. The novel objects enables testing models on out-of-domain. DOCCI was specifically curated to evaluate fine-grained reasoning in image-text models. The image captions cover attributes, spatial relationships, object counts, text rendering, and

world knowledge. These data allow will allow us to evaluate models in their descriptive understanding or generation capabilities. For multi-generation GenEval evaluations (MGG), we employ the `GenEval-R` (GenEval Rewritten) dataset Chen et al. (2025), which extends the short GenEval prompts into long descriptive texts which better match models' outputs.

## 4.1 SEMANTIC DRIFT PROTOCOL FINDINGS

From our evaluations, we observe several interesting qualitative patterns. Fig. 5 illustrates six of such different ways in which unified models lose information under alternating `T2I` ↔ `I2T` cycles: 1. **Position Inconsistency**: the model fails to preserve spatial relationships that are central to the scene, 2. **Object Misidentification**: low-fidelity renderings lead to incorrect re-captioning, 3. **Style Transition**: the model may change the style of an image, particularly for rare object pairings (e.g., a horse on a keyboard), 4. **Quantity Inconsistency**: numerical counts may be inflated, 5. **Object Hallucinations**: new elements are introduced, 6. **Color Inconsistency**: important colors are not retained.

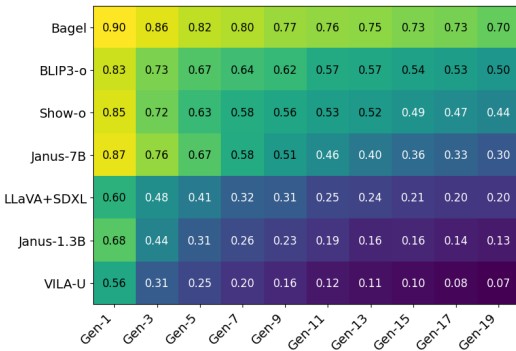

Figure 7: MGG results on the GenEval Rewritten dataset. This heatmap shows the overall performance across the six tasks described in the GenEval Ghosh et al. (2023) benchmark. On average, BAGEL consistently drifts the least from the semantic meaning of the original caption.

Next, we present the empirical results in Fig. 6 which shows the scores obtained from Eq. 1 for all distance mappings, $\{text \rightarrow text, image \rightarrow text, text \rightarrow image, image \rightarrow image\}$. These scores are later used to obtain MCD. In the ideal case, the similarities should remain nearly constant across generations. Instead, as shown in these plots we observe consistent degradation in semantic fidelity, with modality dependent asymmetries. Fig. 6(a) measures the similarity between the original caption and the text generated in Text-First-Chain. Top performing models start with a high similarity ($\sim$ 0.65-0.70), however only BAGEL maintains it relatively well, ending around 0.65. In contrast, models like VILA-U and Janus 1.3B exhibit a much steeper decline, with VILA-U's similarity dropping below 0.40, indicating that its generated texts or images quickly lose connection to the original prompt. Fig. 6(b) and Fig. 6(d) offer a cross-modal perspective, evaluating the text $\rightarrow$ image, and image $\rightarrow$ text respectively. In both scenarios, BAGEL maintains a clear lead, while VILA-U's generations drift so severely that their relevance to the original text becomes minimal at later stages. Across both plots, the overall model ranking at the last step is exactly same. Fig. 6(c) measure visual fidelity by comparing the original image to the generated images at subsequent steps in Image-First-Chain. While the leading models perform similar to prior trends discussed above, we notice Janus 1.3B scoring high in the first generation (0.6), but eventually degrading to a low score in the last generation. Overall, this behavior of models performing well in the first generation, but eventually losing context along the generations is a characteristic not reliably captured by conventional single-pass metrics.

Fig. 7 shows that while initial MGG scores are high, they can mask qualitative differences between models. For instance, BAGEL produces more faithful generations than SHOW-O even with similar initial scores, a divergence that only becomes numerically apparent in later generations as semantic drift occurs. This underscores that cyclic evaluation reveals quality differences that single-pass metrics obscure. Furthermore, performance collapses most dramatically on compositional tasks like positioning and attribute binding (Fig. 11), suggesting this weakness is a key cause of semantic drift. Overall performance, summarized in Fig. 8, plots MGG against $MCD_{avg}$ and reveals a correlation between object-level and embedding-level metrics. A notable exception is the decoupled LLaVA+SDXL system, which scores well on MGG but poorly on MCD, indicating it can render specific objects while failing to preserve holistic scene semantics. Across all evaluations, BAGEL consistently shows the most resilience to semantic drift, likely due to its scale, architecture, and training on diverse interleaved datasets, which makes it uniquely robust against the compounding errors our protocol exposes.

The findings above reveal that semantic drift is not linear, but rather catastrophic. Once a model commits a critical error (e.g., the brown bus turning yellow in Fig. 5), the original semantic meaning is irretrievably lost. While secondary details may continue to degrade in subsequent steps (e.g., the suitcase information in Fig. 5), the metric effectively saturates once the core semantic components are compromised. Hence, the maximum number of generations, $G$, needs to be sufficiently large to allow the drift to manifest, but not so large as to reach information saturation. We anchor $G = 20$ based on the observation that the lowest performing model, VILA-U, reaches a near-zero value in the MGG metric by generation 19 as observed in Fig. 7. This duration is therefore optimal, as it also provides strong correlation with our human evaluations.

## 4.2 HUMAN EVALUATION RESULTS

The dual-section design allowed us to capture cross-consistency. Specifically, if a model received the same fidelity rating (e.g., High) for both the caption and the generated image corresponding to the same $(I, T)$ pair, we considered the model consistent. Conversely, a mismatch in fidelity indicated inconsistency. This approach allowed us to identify not only whether inconsistency exists, but also which type is more prevalent. For example, as shown in Fig. 9 and Appendix 14, most unified models primarily exhibit inconsistencies in the generation task. In Fig. 9(c) BAGEL, shows strong understanding but occasionally fails to generate images with high fidelity.

The ranking component served to compare human-perceived relevance across models. We

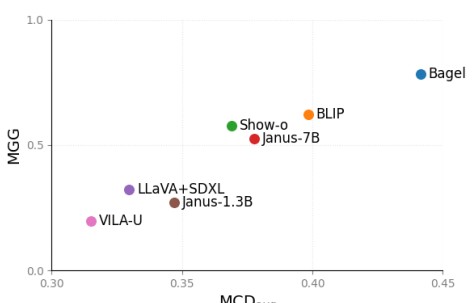

Figure 8: Comparison across MCD and MGG shows that BAGEL achieves the highest performance on both metrics, while VILA-U lags in both. The models align in a linear fashion, hinting at a correlation between the two scores.

computed the mean ranking of each model across all samples to establish a human-based ordering. These rankings were then compared with our embedding-based metrics to assess alignment with human judgment. As shown in Fig.12, there is a clear correlation between human rankings and the MCD metric, validating our embedding-based approach as a reliable proxy for human-perceived semantic consistency.

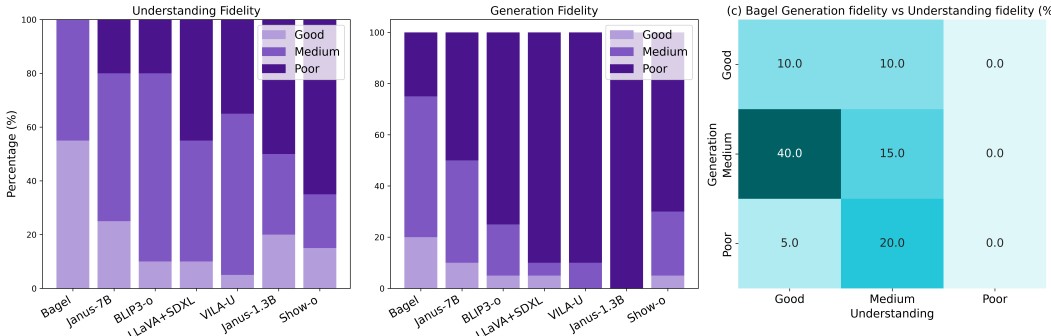

Figure 9: Human evaluation of cross-consistency. First two plots show the percentage of samples (y-axis) rated with a fidelity score (color) for different models. We see that most models gain a high amount of Poor fidelity score in image generation, whereas understanding is pretty balanced, with Bagel almost always getting Medium or better. The third plot illustrates a finer look at the responses for the Bagel model. We see that while Bagel has 10% of Good-understanding-Bad-Generation type of inconsistency, it does not have any other type of inconsistency.

## 5 CONCLUSION

We introduced the Semantic Drift Protocol (`SDP`), a cyclic evaluation framework that alternates image-to-text (`I2T`) and text-to-image (`T2I`) to measure how unified models preserve meaning over repeated modality shifts. By combining embedding-based metrics (`MCD`) and object-level fidelity (`MGG`), SDP exposes vulnerabilities that single-pass evaluations cannot capture. Evaluating seven recent models on the sampled `Nocaps+Docci400` dataset shows substantial variability: BAGEL maintains the strongest cross-modal stability, VILA-U and JANUS variants drift quickly, and Show-o, while not always leading initially, degrades more gracefully across generations. Human evaluations confirm these findings, showing that automated metrics like MCD strongly align with human judgments. These results demonstrate that single-pass benchmarks can overstate robustness, whereas our cyclic evaluation validated by human judgment reveals hidden inconsistencies between image understanding and image generation. We conclude that cyclic evaluation is essential for reliable assessment of unified models.

## REPRODUCIBILITY STATEMENT

All code used to generate images and captions relies on publicly available open-source implementations from the respective GitHub repositories of the models. The evaluation code required to compute the reported scores will be released publicly. All datasets used are publicly available, as referenced in the paper, and no proprietary data was used. Evaluation procedures are fully described in the paper, and the exact code used to compute the reported scores is included with the submission. We believe these details are sufficient for independent researchers to reproduce our results.

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

APPENDIX

This appendix provides additional details and extended analyses that complement the results presented in the main paper. We first describe the models used in our experiments, including their parameterization and image generation settings. We then report further evaluations using CLIP embeddings, and present comprehensive results from the extended multi-generation GenEval benchmark.

## A  MODELS & PARAMETERS

Tab. 1 lists the models included in our evaluations, along with their parameter counts and image resolutions used during generation.

| Name | Parameters | Image Resolution |
|---|---|---|
| BAGEL | 14B - Mixture of Transformers (7B Active) | 1024×1024 |
| Show-o | 1.3B | 512×512 |
| Janus | 1.3B | 1024×1024 |
| Janus Pro | 7B | 1024×1024 |
| VILA-U | 7B | 256×256 |
| Blip-3o | 4B | 1024×1024 |
| LLaVA 1.5 + SDXL | 7B + 3.5B | 1024×1024 |

Table 1: Overview of models used in our experiments, including parameter counts and image resolution. The BAGEL model is a mixture-of-transformers architecture, where 7B parameters are active during inference.

## B  RELATED WORKS

**Unified Models** `T2I` generation has advanced with diffusion-based models such as DALL·E 2 Ramesh et al. (2022), Imagen Saharia et al. (2022), and Stable Diffusion Rombach et al. (2022), which synthesize high-fidelity images from textual prompts. Image captioning, on the other hand, has evolved from CNN-RNN pipelines Shi et al. (2015) to transformer-based decoders Cornia et al. (2020); Liu et al. (2023) trained with large web-scale data. Recent works in unified models have started investigating how to unite understanding and generation under one architecture. Chameleon Team (2025) is one of the early works in this domain which aimed to auto-regressively generate text tokens and image embeddings. Later, Transfusion Zhou et al. (2024) fused the autoregressive and diffusion loss within a single architecture. Show-o Xie et al. (2024) has also used two different objectives, next token prediction for text generation, and masked token prediction Chang et al. (2022) for image generation. Vila-u Wu et al. (2025) uses next token prediction with different text and vision decoders. Janus and Janus-pro Wu et al. (2024) employ separate encoders for image input during understanding and generation. The idea is that a model might require different level of information for understanding and generation. Other works like Blip-3o Chen et al. (2025) demonstrates good quality of image generation by leveraging a separate diffusion transformer head. A recent work, BAGEL Deng et al. (2025) demonstrates some unique capabilities of unified models by training on a large-scale interleaved dataset.

**Prior Evaluations** A variety of benchmarks have been proposed to evaluate the multimodal capabilities of vision-language models. MME Fu et al. (2024) assesses basic perception and reasoning through fine-grained tasks such as object existence, color, and OCR. MMBench Liu et al. (2024) introduces more complex queries, especially in spatial reasoning. MMMU Yue et al. (2024) focuses on college-level academic problems in fields such as science and art. MM-VET Yu et al. (2024) covers diverse skills, including math, OCR, and spatial understanding. MathVista Lu et al. (2024) targets mathematical reasoning in visual contexts such as graphs. MMVP Tong et al. (2024) highlights flaws in existing benchmarks using CLIP-similar but human-atypical images. The FID score

Heusel et al. (2017) provides a metric-based evaluation of image generation quality, while Geneval Ghosh et al. (2023) benchmarks generative vision language models in instruction follow-up and visual grounding. Iterative text-image generation loops have rarely been studied in systematic depth. The work in Bahng et al. (2025) is the closest in spirit where they they use cycle-consistency to create a preference dataset. However, this work only looks at one generation and is limited to VLM models in general and does not consider unified models.

## C  More Results Using CLIP Embeddings

The main paper Fig. 6 presents $S_\delta(g)$ results for text $\rightarrow$ text and image $\rightarrow$ image settings using MPNet (for textual embeddings) and DINO (for visual embeddings). Here, we extend this analysis by incorporating CLIP as an additional backbone, shown in Fig. 10.

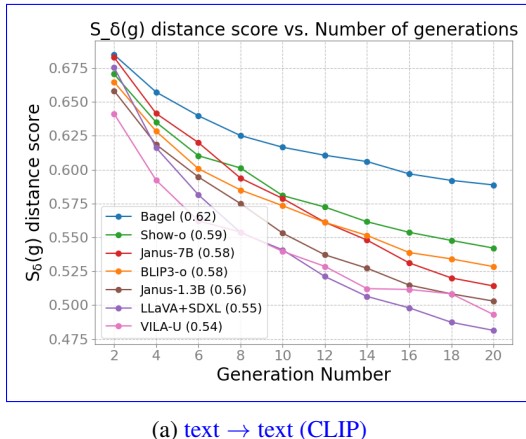
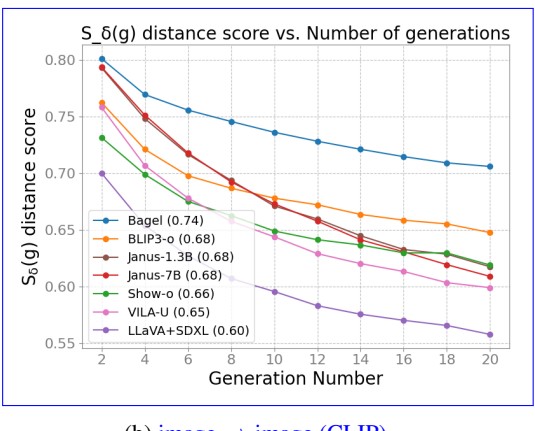

(a) text $\rightarrow$ text (CLIP)  (b) image $\rightarrow$ image (CLIP)

Figure 10: We show $S_\delta(g)$ distance scores computed using CLIP for both text $\rightarrow$ text and image $\rightarrow$ image.

For the Text-First-Chain, **text $\rightarrow$ text** comparison shown in Fig. 10 (a), CLIP similarities are consistently lower than those produced with MPNet as shown in Fig. 6 (a). Despite this, the overall ranking of models is preserved as BAGEL continues to outperform others.

For the Image-First-Chain, $image \rightarrow image$ comparison shown in Fig. 10 (b), the models have higher similarities in the first generation compared to DINO in Fig. 6 (c). The relative order of model performance remains consistent with DINO.

## D  Analysis of Multi-Generation GenEval Results

Fig. 11 shows multi-generation performance in the six tasks from GenEval benchmark. In these heatmaps, darker shades represent lower accuracy. Results from later generations reveal that a model's proficiency in complex tasks is highly susceptible to generational semantic decay, a weakness that single-step evaluations fail to capture.

**Fig. 11(a) Single Object:** The simplest task, requiring generation of a single specified object. Nearly every model achieves near-perfect accuracy in the first generation, but consistency issues appear quickly. VILA-U shows clear degradation, struggling to maintain even one concept.

**Fig. 11(b) Two Objects:** This task assesses handling two entities. The performance drop-off is more pronounced than in the single-object case. Models like Janus 1.3B and LLaVA+SDXL, along with VILA-U lose the ability to consistently generate both objects after only a few generations.

**Fig. 11(c) Counting:** Tests counting capabilities. Initial accuracy is high, but many models fail rapidly, replacing precise numbers (e.g., "three dogs") with vague quantities (e.g., "some dogs"), leading to cascading errors in subsequent generations.

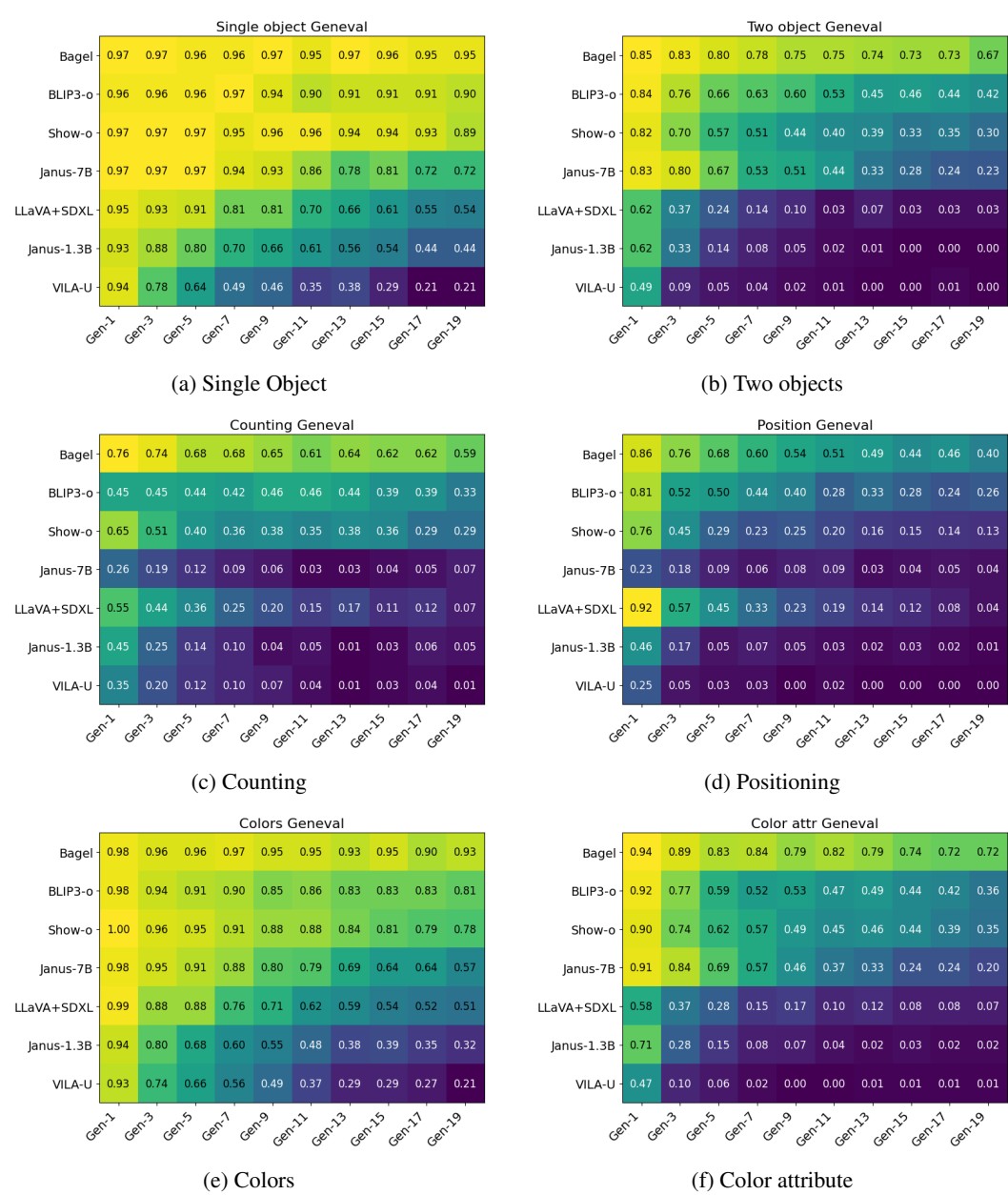

Figure 11: **Detailed Multi-Generation GenEval (`MGG`) Results.** Performance of unified models using `MGG` across 20 generations for six different evaluation categories: (a) Single Object, (b) Two Objects, (c) Counting, (d) Positioning, (e) Colors, and (f) Color Attribute. Darker colors indicate higher accuracy. The results show that while initial performance is high for many models, consistency varies significantly over successive generations, especially for complex tasks.

**Fig. 11(d) Positioning:** Evaluates spatial reasoning (e.g., "a cup to the left of a plate"). Accuracy plummets after the first generation for most models. Preserving spatial relationships proves extremely difficult. BAGEL maintains accuracy longer than other models.

**Fig. 11(e) Colors & 11(f) Color Attribute:** These assess attribute binding. "Colors" is simpler, while "Color Attribute" requires binding colors to specific objects. Both show rapid decay, particularly (f). Models often forget or swap colors. Only top performers retain any meaningful accuracy beyond the initial generations.

# E  CORRELATION OF HUMAN ANALYSIS WITH MCD AND MGG

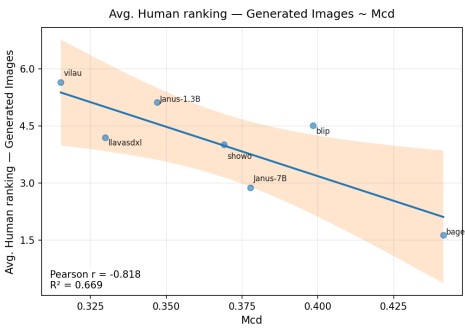 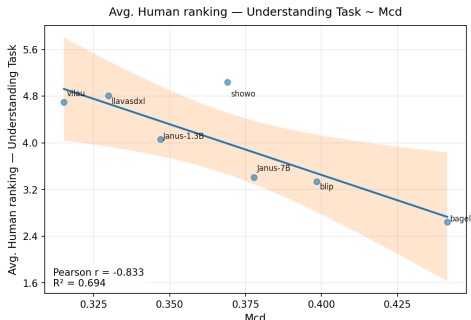

Figure 12: Validation of the $\text{MCD}_{avg}$ metric against human judgments. For both image generation (a) and understanding (b), a lower (better) average human ranking strongly correlates with a higher (less drift) $\text{MCD}_{avg}$ score. This alignment validates that $\text{MCD}_{avg}$ serves as a reliable proxy for human-perceived cross-consistency.

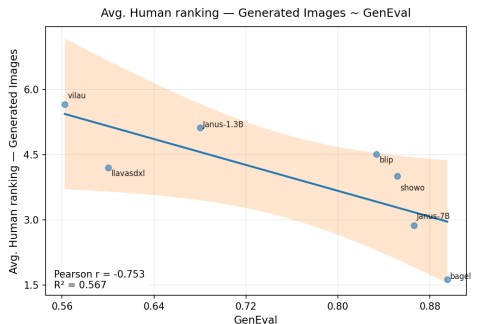 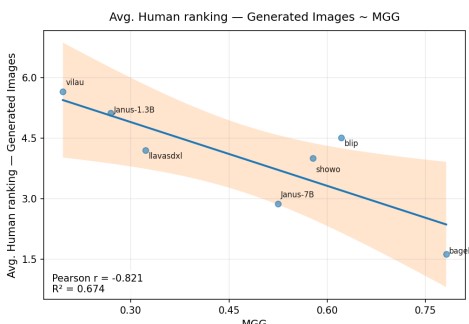

Figure 13: (left) We demonstrate correlation between the GenEval metric against human judgments. (right) We show correlation of MGG agains human judgement. We find our metric correlates more strongly with human perception compared to classic GenEval.

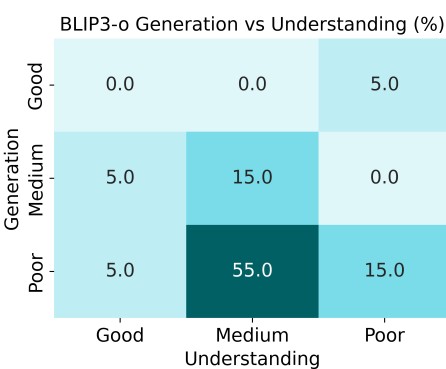

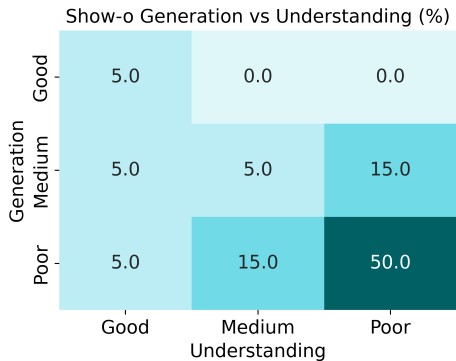

(a) BLIP3-o: Generation vs Understanding fidelity (%)

(b) Show-o: Generation vs Understanding fidelity (%)

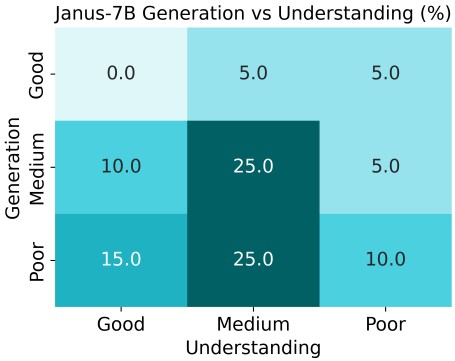

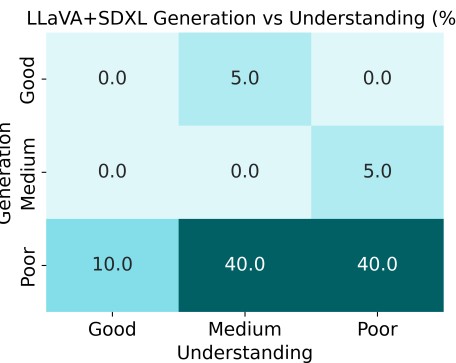

(c) Janus-7B: Generation vs Understanding fidelity (%)

(d) LLaVA+SDXL: Generation vs Understanding fidelity (%)

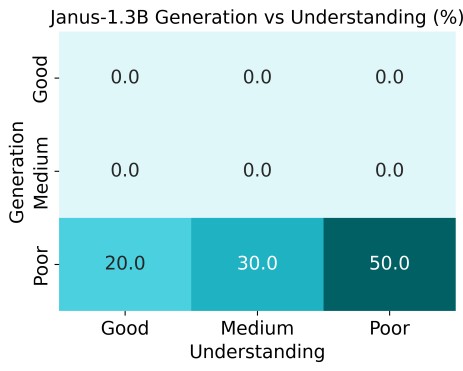

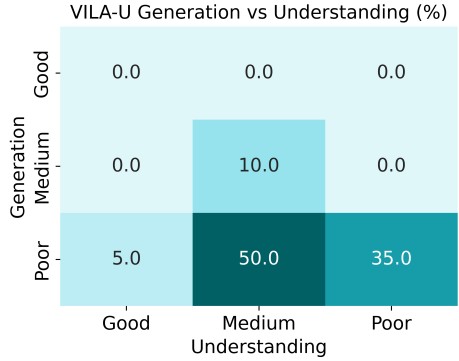

(e) Janus-1.3B: Generation vs Understanding fidelity (%)

(f) VILA-U: Generation vs Understanding fidelity (%)

Figure 14: Heatmaps showing the generation fidelity versus understanding fidelity for six models. Each cell shows the percentage of instances in that category. Darker shades indicate higher percentages. In our definitions, the top right corner and the bottom left corner of each heatmap would identify the two types of inconsistencies. As we can observe it is more likely that given a concept a model would be able to understand well while generating poorly rather than the opposite.

# F  SEMANTIC DRIFT EXAMPLES

The following figures illustrate how semantic drift can occur due to hallucination in either generation or understanding.

| Generation 0 (input) | Generation 2 | Generation 4 | Generation 6 | Generation 8 |
|---|---|---|---|---|
| A clear photo of a black vase placed on a simple flat surface. The vase has a sleek and elegant design, with a glossy finish that reflects light subtly. Its shape is classic, featuring a flared opening at the top. The background is plain and unobtrusive, ensuring that the viewer's attention is drawn entirely to the vase. | The image features a black vase with a glossy finish. The vase has a smooth, rounded body that tapers slightly towards the top. It has a flared opening at the top, which is wider than the base. The vase is placed on a flat, light-colored surface, and the background is a plain, light grey color, providing a neutral backdrop that highlights the vase. The lighting in the image is soft, casting subtle shadows and reflections on the surface of the vase, which enhances its sleek and polished appearance. The overall composition is simple and minimalist, focusing solely on the vase without any additional objects or distractions. | The image displays a black vase with a glossy finish. The vase has a bulbous body that tapers slightly towards the neck, which then widens into a flared opening. The surface of the vase is smooth and reflective, showing a slight reflection of the surrounding environment. The vase is placed on a flat, light-colored surface, and the background is a plain, light grey color, providing a neutral backdrop that highlights the vase. The lighting in the image is soft and even, casting minimal shadows and emphasizing the vase's sleek and modern design. There are no texts or additional objects in the image. The style of the image is a straightforward product photograph, likely intended to showcase the vase's design and color. | The image displays a black vase with a glossy finish. The vase has a spherical body that tapers slightly towards the top, forming a narrow neck that widens into a flared opening. The surface of the vase is smooth and reflective, indicating a high-quality finish. The vase is placed on a flat, light-colored surface, and the background is a plain, light grey, which contrasts with the black color of the vase, making it stand out prominently. The lighting in the image is soft and even, casting subtle shadows that enhance the three-dimensional appearance of the vase. The overall composition is simple and minimalist, focusing solely on the vase without any additional objects or distractions. | The image features a black vase with a glossy finish. The vase has a rounded body that tapers slightly towards the top, forming a flared opening. The surface of the vase is highly reflective, capturing the light and creating subtle highlights and shadows. The vase is placed on a flat, light-colored surface, possibly a table or countertop, against a plain, light grey background. The overall composition is simple and minimalist, focusing on the sleek and modern design of the vase. |
| **Generation 1** | **Generation 3** | **Generation 5** | **Generation 7** | **Generation 9** |

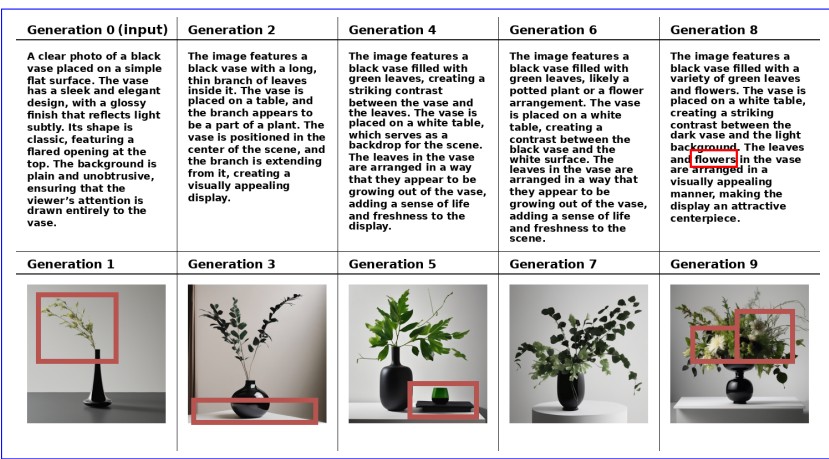

(a) Bagel is able to maintain the core subject of the topic across generations.

| Generation 0 (input) | Generation 2 | Generation 4 | Generation 6 | Generation 8 |
|---|---|---|---|---|
| A clear photo of a black vase placed on a simple flat surface. The vase has a sleek and elegant design, with a glossy finish that reflects light subtly. Its shape is classic, featuring a flared opening at the top. The background is plain and unobtrusive, ensuring that the viewer's attention is drawn entirely to the vase. | The image features a black vase with a long, thin branch of leaves inside it. The vase is placed on a table, and the branch appears to be a part of a plant. The vase is positioned in the center of the scene, and the branch is extending from it, creating a visually appealing display. | The image features a black vase filled with green leaves, creating a striking contrast between the vase and the leaves. The vase is placed on a white table, which serves as a backdrop for the scene. The leaves in the vase are arranged in a way that they appear to be growing out of the vase, adding a sense of life and freshness to the display. | The image features a black vase filled with green leaves, likely a potted plant or a flower arrangement. The vase is placed on a white table, creating a contrast between the black vase and the white surface. The leaves in the vase are arranged in a way that they appear to be growing out of the vase, adding a sense of life and freshness to the scene. | The image features a black vase filled with a variety of green leaves and flowers. The vase is placed on a white table, creating a striking contrast between the dark vase and the light background. The leaves and flowers in the vase are arranged in a visually appealing manner, making the display an attractive centerpiece. |
| **Generation 1** | **Generation 3** | **Generation 5** | **Generation 7** | **Generation 9** |

(b) LLaVA+SDXL hallucinates new elements across generations, inducing drift.

Figure 15: Example showing two text-first chains: Bagel vs LLaVASDXL. In this example, Bagel keeps the semantic meaning "a black vase" across the generations, whereas, LLaVA+SDXL continuously hallucinates new objects or modifications mainly during its generation task.

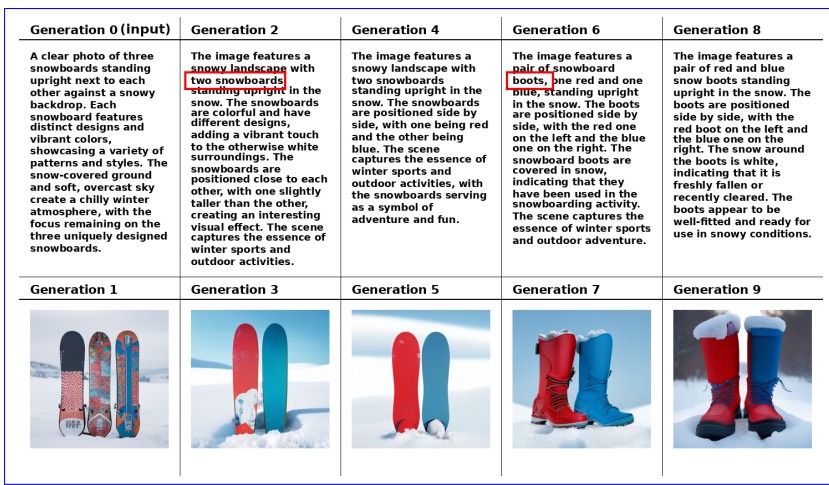

(a) Bagel's output remains similar across modalities across generations. The idea of three snowboards stay stable over the generations.

(b) Show-o misunderstands its own outputs, increasing drift. It first misunderstands three boards as two (generation 3), and then misunderstands the boards as boots.

Figure 16: Example showing two text-first chains: Bagel vs Show-o. Bagel is consistent, where Show-o shows semantic drift mainly due to poor understanding during the understanding task.

## G  EXAMPLES OF CROSS-INCONSISTENCIES

Figures 17 and 18 illustrate two distinct types of inconsistencies observed in unified models. In Figure 17, when the BLIP3-o model processes an image of a worn-down rocky stairway (I2T), it hallucinates a stream of water. Yet, when given the corresponding text prompt (T2I), it generates an accurate image. In contrast, Figure 18 shows the reverse: the BAGEL model correctly describes the image of a dog (I2T), but the input text prompt (T2I) produces a low quality depiction of the animal. These cases suggest that models perform better at generation for some concepts and at understanding for others.

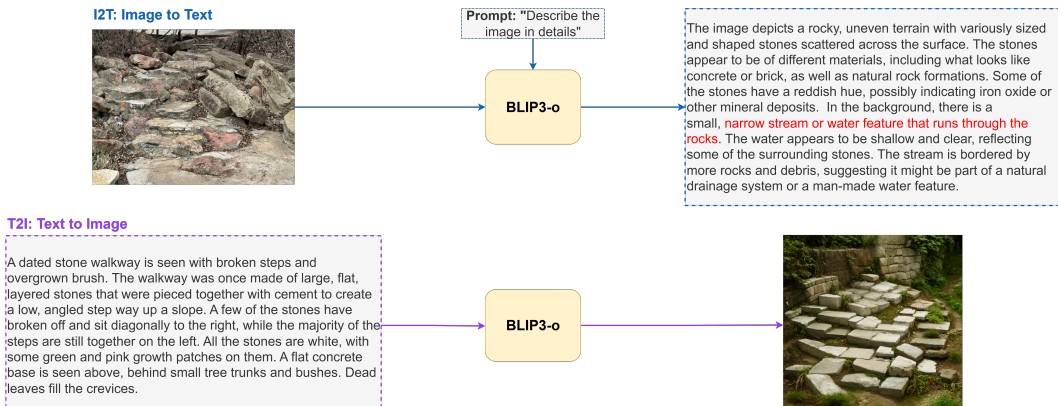

Figure 17: An example of cross inconsistency in the BLIP3-o unified model. When the model is prompted (I2T) to describe the given image (top), it incorrectly mentions a water stream in the image. However, when BLIP3-o is given the text prompt (T2I) (bottom), it generates an image that correctly represents the prompt. This exposes a UM inconsistency: BLIP3-o's correct generation (T2I) does not carry over to visual reasoning (I2T) for this concept.

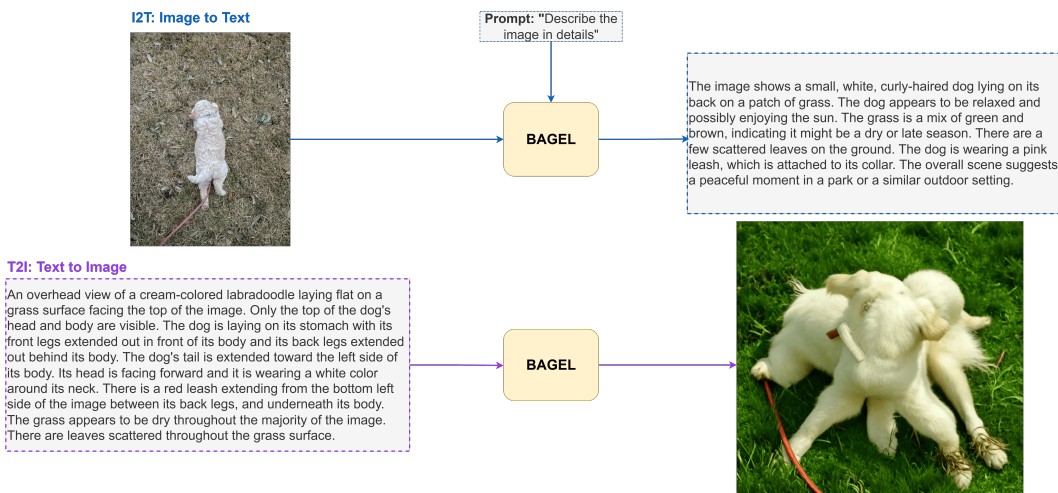

Figure 18: Another example of cross inconsistency in the BAGEL unified model. When the model is prompted (I2T) to describe the given image (top), it correctly describes the dog and its surroundings. However, when BAGEL is given the text prompt (T2I) (bottom), it fails to generate the animal. In this case, BAGEL's correct understanding (I2T) does not carry over to generation (T2I).

