# OpenReview forum: "The Telephone Game: Evaluating Semantic Drift in Unified Models"
_ICLR.cc/2026/Conference — Submitted to ICLR 2026_

### Official Review · Reviewer_MboD · 2025-10-30

**Soundness:** 2
**Presentation:** 2
**Contribution:** 2
**Rating:** 4
**Confidence:** 3

**Summary:**

The paper proposes an evaluation benchmark for unified models, termed the Semantic Drift Protocol (SDP), which focuses on measuring semantic drift during the image-to-text (I2T) and text-to-image (T2I) generation cycle. Specifically, the authors introduce two metrics:  (i) Mean Cumulative Drift (MCD), an embedding-based measure of overall semantic loss; and  (ii) Multi-Generation GenEval (MGG), an object-level compliance score that extends GenEval.  Extensive experiments demonstrate the effectiveness of the proposed evaluation framework.

**Strengths:**

* SDP takes multi-turn generation tasks into account, addressing a gap in previous evaluation frameworks.
* The experimental design is thorough, and the results yield insights with meaningful implications for the research community.

**Weaknesses:**

* The proposed framework lacks novelty. The main contributions, MCD and MGG, are relatively straightforward: MCD is essentially the mean of multi-turn embedding similarity scores, while MGG is a direct extension of GenEval.
* The motivation is somewhat unclear. Although multi-turn dialogue tasks for unified models represent an important research direction, the proposed evaluation framework does not appear to align well with realistic application scenarios. In most real-world settings, after obtaining an initial dialogue output, users would typically modify their prompts to refine the result rather than repeatedly generating the same (I, T) pair as described in this paper.
* While the paper introduces an evaluation framework, it would be more compelling if the authors demonstrated its practical utility, for instance, by performing post-training or fine-tuning based on the proposed scores and showing measurable improvements in model performance.
* The paper includes human evaluation results but provides insufficient details about the evaluation procedure. A more comprehensive description of the human evaluation protocol is necessary to ensure the fairness and reliability of the results.
* The writing lacks clarity and coherence, and the paper contains numerous typographical errors (e.g., line 25: “image-totext” should be “image-to-text”).

**Questions:**

See 'Weakness'.

---

> ### Author Response · Authors · 2025-11-24
>
> Thanks to the reviewer for acknowledging the research gap addressed in this work, and the thoroughness of the experimental design.
>
> > Weakness 1: MCD and MGG are Relatively Straightforward
>
> We thank the reviewer for raising this point. We emphasize that the novelty of our work in introducing the first evaluation protocol that measures two fundamental properties of unified models that existing metrics simply do not capture: Semantic Drift and Cross-modal Consistency.
>
> Current metrics assess understanding or generation independently, and therefore cannot capture whether a unified model maintains coherent, shared semantics across modalities. Our multi-generation cyclic protocol is new: it jointly stresses both I2T and T2I, revealing error accumulation and cross-modal inconsistencies that single-pass evaluations miss.
> This leads to practical, previously unobserved insights as shown in Fig. 2,6,7,15,16.
> For example, VILA-U performs well under isolated evaluations but rapidly degrades under our proposed multi-generation protocol (Fig. 6, 7). Likewise, standard GenEval shows saturation across unified models (e.g., Janus-7B vs. BAGEL), whereas our MGG metric provides a clearer separation and shows stronger correlation with human judgments than GenEval (Fig. 16). These results demonstrate that current benchmarks miss critical weaknesses in unified models, and that drift and cross-modal consistency must be measured directly.
>
> Finally, our protocol offers a holistic design, combining text–text, image–image, and cross-modal embedding drift with object-, count-, spatial-, color-, and relation-level fidelity (MGG). This approach allows us to evaluate unified models comprehensively and uncover failure modes that are invisible to existing evaluation practices.
>
> > Weakness 2: Evaluation Setup does not Reflect Realistic Application Scenarios
>
> The two tasks we target: image captioning (I2T) and image generation (T2I) are themselves standard, widely deployed capabilities of unified models. These are precisely the tasks used in our human evaluation, which reflects genuine real-world usage.
>
> Keeping the starting input fixed allows us to isolate and measure model-intrinsic drift, rather than drift introduced by user-driven prompt changes. By holding the input constant, the protocol ensures that any deviation in meaning arises directly from the model’s own I2T↔T2I transformations.
>
> Our analysis highlights that
> - In Fig. 16(b), the Show-o model gradually drifts from three snowboards to a pair of boots across generations.
> - In Fig. 15(b), drift appears as early as Gen-1, demonstrating that some models exhibit instability immediately, which aligns strongly with our human evaluation findings.
>
> These inconsistencies are also captured in both MGG and human ratings, highlighting their practical relevance to real captioning and generation scenarios.
>
> We further want to emphasize that internal cross-consistency of Unified Models itself is an important scientific question which we explore.
>
> > Weakness 3: Lack of Practical Utility
>
> This is the first kind of work to evaluate Semantic Drift and Cross-Modal Consistency, two properties that are fundamental to unified models yet unmeasured by existing metrics. We demonstrate the practical utility of this evaluation by showing strong alignment with real-world image captioning and image generation performance in our human study.
>
> While using drift metrics for fine-tuning is indeed a compelling direction, we consider this a separate future research problem. Our contribution here is to establish an evaluation protocol that reveals behaviors not visible through current metrics and lays the groundwork for such future optimization methods.

---

> ### Author Response · Authors · 2025-11-24
>
> > Weakness 4: Insufficient Details About the Human Evaluation
>
> We appreciate the feedback. As per your suggestion, we have expanded Section 3.3 to provide clearer and more complete details of our human evaluation protocol.
>
> 1. We sampled 100 Text-First and 100 Image-First chains from the MCD evaluation set, resulting in 200 evaluation examples.
> 2. Six human annotators participated. Each annotator received a comparable number of samples, and every example was annotated independently by two different annotators.
> 3. Annotators used a web interface to provide two judgments per sample:
>     a. three-level fidelity score (Good / Medium / Poor), and
>     b. ranking of model outputs by semantic correctness relative to the original input.
> 4. To prevent model-specific or positional bias, we masked model identities and randomized the output order for every evaluation instance.
> 5. Rather than averaging annotators’ opinions, we treat each annotation as an independent data point, allowing us to measure consistency across annotators without collapsing their perspectives.
>
> These details are now explicitly included in the main text.
>
> > Weakness 5: Lack of Writing Clarity
>
> We have addressed the mentioned typographical errors, and improved writing clarity:
> 1. Grammatical flow in the overall paper (Edits are shown in blue font)
> 2. Human study explained in more details (Section 3.3)
> 3. Added details on Drift (Section 4.1)

---

> > ### Comment · Reviewer_MboD · 2025-11-27
> >
> > Thanks to the authors for clarifying the questions I raised. Below are my follow-up comments.
> >
> > The reviewer has previously used Bagel, which supports high-quality multi-round image editing. Specifically:
> > 1. User: Prompt
> > 2. Bagel: Image
> > 3. User: Editing instruction
> > 4. Bagel: Edited image
> > 5. User: Editing instruction
> > 6. Bagel: Further edited image
> > 7. ...
> >
> > Throughout this multi-round editing process, there is no obvious semantic drift. Does this imply that Bagel achieves a strong alignment between generation and understanding? Furthermore, compared with this editing-style multi-turn workflow, what are the advantages of the proposed method that focuses on multi-round generation with consistent semantics?

---

> > > ### Author Response · Authors · 2025-12-03
> > >
> > > Thank you for your thoughtful follow-up. Bagel's performance in multi-turn editing is a great observation, and here we share our perspective.
> > >
> > > - **Comparison between multi-round editing and cross-modality alignment:**
> > > Image editing and cross-modal consistency behaviors are mechanistically similar as they both require to preserve the core content of the input when generating an image. We can formalize image editing as an `Image + Text -> Image` task, where the model must interpret an instruction relative to an existing visual state to produce a new one. Similarly, the chains in our Semantic Drift Protocol contains multiple `Text -> Image` transformations where the image must reflect the input text. Consequently, information retention is critical for both; a failure manifests as "instruction ignoring" in editing or "semantic decay" in our protocol. Bagel demonstrates superior performance in our benchmarks, maintaining the highest stability in embedding distance (MCD) and object compliance (MGG) across 20 generations which correlates directly with its strong editing scores on GEdit-Bench and Intelligent Bench. This indicates that the ability to faithfully preserve semantics during modality transitions is effectively measured in our metric.
> > >
> > >
> > > - **Advantages of proposed method:**
> > > The advantage of the proposed Semantic Drift Protocol (SDP) over the editing-style multi-turn workflow is its focus on measuring how well aligned a model's image understanding and generation tasks are. While Bagel is robust in editing, our protocol reveals that it can still drift when forced to reconstruct concepts from text alone. For example Bagel fails to generate a chessboard in a winning condition (Figure 2) that matches its own correct textual description, or failing to generate top view image of a dog despite a highly descriptive caption. Existing editing benchmarks like GEdit-Bench, and Intelligent Bench primarily test a model's ability to modify an image conditioned on both text and the visual tokens of the preceding image. However, a model may be good at generating high-quality images conditioned on previous text and images without being equally capable to describe images on the same topic (or vice-versa). The SDP addresses this by explicitly evaluating both generation and understanding in a singular evaluation protocol.

---

### Official Review · Reviewer_iUSY · 2025-10-30

**Soundness:** 2
**Presentation:** 3
**Contribution:** 2
**Rating:** 4
**Confidence:** 3

**Summary:**

This paper proposes the Semantic Drift Protocol, an evaluation framework for unified multimodal models that perform both text-to-image and image-to-text tasks. It introduces two key metrics — Mean Cumulative Drift and Multi-Generation GenEval — to measure how well models preserve meaning across multiple modality conversions, addressing a gap in existing benchmarks that only evaluate single-pass performance. The authors also present a new benchmark dataset, NoCaps + DOCCI400, and evaluate seven recent models, finding large differences in their cross-modal semantic stability.

**Strengths:**

Most existing multimodal benchmarks assess text-to-image and image-to-text performance separately, giving little attention to semantic consistency across iterative modality exchanges.  Such consistency is especially important for tasks like multi-turn visual question answering, vision–language dialogue, video captioning, and related multimodal tasks, where meaning must remain stable across modalities.

The proposed evaluation protocol fills this gap by simulating successive T2I–I2T transformations, capturing how meaning shifts across repeated modality exchanges — a property overlooked by single-pass metrics.

By comparing shared-weight and decoupled architectures, the paper further shows that shared models tend to preserve meaning more effectively across rounds.

**Weaknesses:**

The paper presents an interesting idea and addresses a gap in multimodal evaluation, but its conceptual and empirical grounding are weak.
The definition of *semantic drift* lacks rigor, the metrics suffer from representational bias, and the experiments fail to validate the stated motivation — that higher semantic consistency benefits downstream multimodal reasoning tasks.
Overall, the work remains exploratory rather than a robust or generalizable benchmark.

1. Conceptual Ambiguity in the Definition of Semantic Drift
The central concept is poorly defined. Treating semantic drift as embedding similarity decay is conceptually weak and conflates representation change with genuine semantic degradation. This assumption is unverified—embedding distance measures variation in representation, not meaning. As a result, benign paraphrasing or stylistic variation can be incorrectly penalized as drift, while the metric itself merges object-, attribute-, and relation-level shifts into a single opaque score.

 2. Bias from Embedding-Space Alignment
Both metrics (MCD and MGG) rely on pretrained embeddings and detectors, introducing representational bias.  Models trained in similar embedding spaces may appear more consistent—not due to true semantic stability, but because their representations align with the evaluation backbone.
Without checks using alternative embeddings or detectors, it is unclear whether the reported rankings reflect genuine stability or embedding overlap, undermining the claim that SDP is architecture-agnostic.

3. Lack of Downstream or Practical Validation
All experiments are limited to synthetic T2I–I2T cycling and a human-correlation study, with no evidence that higher semantic consistency improves real multimodal tasks such as VQA, captioning, or visual dialogue. Without quantitative or qualitative downstream validation, the claimed importance of the proposed metrics remains speculative, and it is unclear whether the evaluation protocol reflects meaningful model behavior beyond the test loop.

**Questions:**

Definition of Semantic Drift
The definition of “semantic drift” as embedding similarity decay is ambiguous. It does not distinguish genuine semantic loss (e.g., object or relation errors) from benign changes like paraphrasing or stylistic variation. How can the authors ensure that their metric reflects semantic degradation rather than surface variation?

Embedding Dependence
Since both MCD and MGG rely on pretrained embeddings (CLIP, DINO, MPNet), results may favor models trained in similar representation spaces. Have the authors tested whether model rankings remain consistent under alternative embeddings or random projections?

Downstream Relevance
The paper claims that higher semantic consistency benefits multimodal task, but provides no downstream validation. Can the authors show whether SDP scores correlate with task performance on VQA, captioning, or dialogue?

---

> ### Author Response · Authors · 2025-11-24
>
> We thank the reviewer for their time and feedback. We are encouraged that you, along with the other reviewers, agree that Semantic Drift Protocol fills a vital gap in existing benchmarks, which typically assess text-to-image and image-to-text performance in isolation.
>
> > Question 1: Semantic Degradation rather than Surface Variation
>
> We thank the reviewer for raising this insightful question.
> We agree that the term semantic can be overly broad, and we clarify in the revision that our use of “semantic drift” refers to changes in the core semantic content (e.g: objects count, color, attribute relations, spatial position captured by Multi-Generation GenEval metric (MGG)) that occur when a model repeatedly applies its own I2T↔T2I transformations.
> Importantly, Semantic Drift Protocol (SDP) is not based on a single embedding space. It uses multiple complementary signals: MPNet (text→text), DINO (image→image), CLIP (cross-modal), and the MGG metric for object-level fidelity. MGG is particularly useful when object semantics matter more, it relies on object detection, disregards paraphrasing, and focuses directly on whether entities, attributes, and relations are preserved. It can be inspected as an overall score or via its objects count, color, attribute relations, or spatial position components (Fig. 11).
>
> We also observe a strong correlation between MCD and human rankings (Fig. 13), demonstrating that the drift metrics capture meaningful semantic degradation rather than superficial rephrasing.
>
> > Question 2: Bias from Embedding-Space Alignment
>
> Our protocol mitigates representation bias in three ways.
>
> First, our embedding based metric is not tied to a single embedding space: we use MPNet (text–text), DINO (image–image), and CLIP (cross-modal), which differ in architecture and training data.
>
> Second, we explicitly compare two Show-o variants: one trained with CLIP features and one without. We show that the MAGViT2 model shows consistent improved scores on 4 different settings. Because MPNet is independent from CLIP, and Show-o does not leverage SigLIP, this consistent result indicates true semantic stability rather than representational alignment.
>
> Third, the MGG metric provides a detector-based score focused on objects, attributes, and relations. MGG yields model orderings consistent with MCD (Fig. 8), further reducing the likelihood of backbone-driven artifacts.
>
> Finally, human rankings correlate with both MCD and MGG (Fig. 12, 13), confirming that the drift we measure reflects meaningful semantic degradation rather than embedding bias.
>
> | Eval Type            | text-to-text (clip) | text-to-text (siglip) | text-to-text (mpnet) | text-to-image (clip) |
> |----------------------|---------------------|------------------------|-----------------------|-----------------------|
> | Show-o (w/ CLIP)     | 0.5875              | 0.3962                 | 0.5749                | 0.2700                |
> | Show-o (w/ MAGViT2)  | **0.5978**              | **0.4040**                 | **0.5957**                | **0.2768**            |
>
> > Question 3: Downstream Relevance
>
> We target two tasks, image captioning (I2T) and image generation (T2I) in both our metrics and in the human evaluation. These two tasks are themselves standard, downstream tasks of unified models. By holding the input constant, and repeatedly performing I2T↔T2I transformations, the protocol ensures that any deviation in meaning arises directly from the model’s own. Importantly, we find that existing automated I2T metrics often rank several models similarly, yet human ratings reveal clear semantic differences, including hallucinations, missing objects, or incorrect attributes, that current isolated metrics fail to capture. SDP therefore surfaces shortcomings that conventional evaluation cannot detect.
>
> For example, single-pass GenEval suggests that Janus-7B performs similarly to BAGEL (Fig. 7), but MGG, MCD, and human evaluation show BAGEL to be substantially more stable and cross-consistent. This demonstrates why multi-generation evaluation is needed.
>
> We further emphasize that our work on cross-consistency and semantic drift explores a scientific aspect of Unified Models that was previously unaddressed.

---

### Official Review · Reviewer_KpeP · 2025-11-01

**Soundness:** 3
**Presentation:** 3
**Contribution:** 3
**Rating:** 8
**Confidence:** 4

**Summary:**

Evaluation of unified multimodal models remains limited, as image understanding and generation are evaluated in isolation. This paper formulates the semantic drift and cross-consistency problems, and shows that existing single-pass metrics cannot thoroughly measure the gap between understanding and generation capabilities. To address this, the paper proposes the Semantic Drift Protocol (SDP), a cyclic evaluation framework that measures how well a unified model preserves semantic fidelity by alternating between image-to-text (I2T) and text-to-image (T2I) generation. They also introduce a new benchmark (Nocaps+Docci400) evaluating seven recent models.

**Strengths:**

* The proposed evaluation protocol effectively measures a unified model's ability to preserve semantic fidelity across modalities, which existing metrics or benchmarks fail to capture. The results in Figure 5 clearly demonstrate this point, providing intriguing evidence of how semantic drift can occur across different categories.
* The main figures (Figures 1 and 2) clearly explain the core problem the paper is addressing.
* The categorization of unified models in Section 2 is clear and well-motivated, and the experiments thoughtfully include models from different categories.
* They provide a human study that confirms a strong correlation between the proposed metric and human-perceived quality.

**Weaknesses:**

* Including a guideline on the recommended number of evaluation cycles would improve reproducibility and facilitate adoption of this protocol in future research.
* It would be helpful to improve the clarity of the x-axis and y-axis labels in Figure 6 (e.g., Sδ(g) distance score vs. Number of generations). Also, the figure should be referenced accurately in the main text (line 398: change "Plot" to "Figure" 6).
* Typos in line 252.

**Questions:**

* It is very interesting that BLIP-3o, a partially shared model, performs exceptionally well on this metric, achieving the second-best result. Do the authors have any insights into why this might be the case?
* Overall, the paper suggests a novel evaluation protocol that can effectively measure a qualitatively different aspect of unified models. Minor presentation quality (e.g., titles, axis labels, and font size of plots, and typos) could be improved for greater clarity and impact.

---

> ### Author Response · Authors · 2025-11-24
>
> Thank you for your thorough review and recognizing that the proposed evaluation protocol effectively addresses semantic fidelity across modalities.
>
> > BLIP3-o's Performance
>
> We believe BLIP-3o’s strong performance arises from the fact that it does not rely on a single shared model for both understanding and generation. Instead, BLIP-3o uses dedicated *extensively pre-trained expert* components: Qwen2.5-VL exclusively for image understanding (I2T), and Lumina-Next DiT exclusively for image generation (T2I). This architectural separation allows Qwen2.5-VL to retain strong comprehension capability. In contrast, fully shared-weight unified models must learn both tasks within a single model, which can lead to task interference and error amplification across cyclic I↔T transformations, precisely what the SDP protocol is designed to surface. On the generation side, Lumina-Next’s diffusion/flow-matching training on CLIP-semantic features is inherently more robust to mild paraphrasing or caption variations than autoregressive decoders, making its T2I outputs more stable under the small linguistic drift that naturally arises in early generations.
>
> This architectural advantage is reflected in the MGG heatmap (Fig. 7). At Gen-1, unified models like BAGEL, Janus, and Show-o achieve higher scores than BLIP-3o. However, their performance deteriorates much more rapidly across generations, while BLIP-3o maintains comparatively stable scores over time. The combination of (i) a strong frozen understanding model, (ii) a dedicated robust diffusion-based generator, and (iii) limited parameter sharing therefore reduces long-horizon semantic drift, explaining BLIP-3o’s favorable performance in our cyclic evaluation.
>
> > Writing Improvements
>
> We have corrected the typo mentioned in line 252, improved the clarity of the x-axis and y-axis labels in Figure 6, and made sure that the figure is referenced accurately in the main text. We have also made further writing improvements in the overall text.
>
> > Recommended Number of Evaluation Cycles
>
> We agree with the importance of inclusion of a guideline, which we now discuss at the end of section 4.1 (SEMANTIC DRIFT PROTOCOL FINDINGS).
>
> From the MCD and MGG graphs (Fig. 6, Fig 7), we can conclude that drift in not linear, it is catastrophic. Once a model makes a critical error, (e.g., hallucinating a plant in a vase (Fig. 15), the original semantic meaning is irretrievably lost.
>
> Often a model requires a few generations to show meaningful drift. Hence, G needs to be big enough to give the models time to show its inconsistencies. On the other hand a very large G, would create saturation of error: once a model loses all its key information, measuring it further adds no new information to the metric.
>
> In the MGG metric, VILA-U model reaches close to 0 (0.07) in Gen-19 (Fig. 7). This can be interpreted as complete semantic drift in terms of the GenEval categories (object count, color, position, attributes, etc.). Hence, we use this model as an anchor to choose the G = 20. This point aligns with our qualitative observations where we see good models being able to retain key information until G=20, while worse models get enough time to shift within this time.

---

### Author Response · Authors · 2025-12-03
**Summary for Area Chair**

Dear Area Chair,

Thank you for your time and effort in handling this submission. We briefly summarize the key points from the rebuttal phase.

## Overall Scores
| Reviewer | Score | Confidence |
|-----------|--------|-------------|
| Reviewer (KpeP)| 8 | 4 |
| Reviewer (iUSY)| 4 | 3 |
| Reviewer (MboD)| 4 | 3 |

## Discussion Summary

Reviewers consistently noted that the paper addresses a significant research gap and that it provides an insightful evaluation of semantic drift in unified models.

### Reviewer KpeP

- Recognized effectiveness of our evaluation protocol.

- *Concerns Resolution*: We discussed why Blip3-o performs well, and provided reasoning on the appropriate number of generation steps.

### Reviewer iUSY

- Agrees with research gap addressed in the paper.

- *Concerns Resolution*: Following reviewer's concerns on possible embedding bias we have added multi-embedding and model-variant analysis. We also clarified the definitions and motivations.

### Reviewer MboD

- Mentions that the work has meaningful implications for the research community.

- *Concerns resolution*: We clarified our metric choices and human study protocol. The **reviewer thanked us for clarifying their initial concerns** and raised a follow-up on Bagel’s multi-round editing, which we responded to.

---

### Meta-Review · Area_Chair_wYdC · 2026-01-05

**Summary:**

This paper proposes an evaluation method for unified models. Specifically, the author introduces the Semantic Drift Protocol (SDP), a cyclic evaluation protocol that alternates I2T and T2I over multiple generations to quantify semantic drift with two metrics, i.e., Mean Cumulative Drift (MCD) and Multi-Generation GenEval (MGG).

The reviewers offer mixed assessments. Reviewer KpeP recognizes the effectiveness of the evaluation protocol and recommends acceptance with a score of 8. Reviewer iUSY with a score of 4 argues that the conceptual and empirical grounding is weak. Similarly, Reviewer MboD with a score of 4, questions the novelty.

After carefully review the paper, all reviewer comments and the rebuttal, the AC agrees that this paper presents an interesting concept and identifies a valid problem. However, the AC agrees with the concerns raised by reviewer iUSY and MboD that the work is currently exploratory and the evaluation method is technically straightforward. Overall, the technical contribution is not strong for acceptance. The AC therefore recommends rejection.

**Reviewer Concerns:**

The concerns raised by Reviewer KpeP have been fully addressed.

The reviewer iUSY's fundamental criticism that the work remains exploratory remains outstanding.

Reviewer MboD: The authors address the questions regarding the evaluation setup and practical utility. However, the primary concern regarding technical novelty remains outstanding.

**Reviewer Scores:**

The fundamental concerns raised by the reviewer iUSY (the work is exploratory) and MboD (lacks technical novelty) have not been adequately addressed by the rebuttal. It is unlikely that either reviewer would raise their score.

---

### Decision · Program_Chairs · 2026-01-26

Reject